# Cortisol levels, heart rate, and autonomic responses in horses during repeated road transport with differently conditioned trucks in a tropical environment

Siengsaw Lertratanachai[1], Chanoknun Poochipakorn[2,3‡], Kanokpan Sanigavatee[2,3‡], Onjira Huangsaksri[2,3‡], Thita Wonghanchao[2,3‡], Ponlakrit Charoenchanikran[4], Chaipat Lawsirirat[1], Metha Chanda[2,3]*

1 Faculty of Sports Science, Chulalongkorn University, Bangkok, Thailand, 2 Department of Large Animal and Wildlife Clinical Science, Faculty of Veterinary Medicine, Kasetsart University Kamphaeng Saen Campus, Nakhon Pathom, Thailand, 3 Thailand Equestrian Federation, Sports Authority of Thailand, Bangkok, Thailand, 4 29th Cavalry Squadron Royal Horse Guard, King's Guard, 2nd Cavalry Brigade, Royal Thai Army, Bangkok, Thailand

☯ These authors contributed equally to this work.
‡ CP, KS, OH and TW also contributed equally to this work.
* fvetmtcd@ku.ac.th

## Abstract

Horse's stress responses have been reported during road transport in temperate but not tropical environments. Therefore, this study measured cortisol levels, heart rate (HR), and heart rate variability (HRV) in horses during medium-distance road transport with different truck conditions in a tropical environment. Six horses were repeatedly transported in either air-conditioned trucks with full (ATF) or space (ATS) loads or non-air-conditioned trucks with full (N-ATF) and space (N-ATS) loads. Blood cortisol was determined beforehand and 5, 30, and 90 minutes post-transport. HR and HRV were assessed pre-transport and at 15-minute intervals until 90 minutes post-transport. Cortisol levels increased significantly in N-ATS horses (but non-significantly in ATF, ATS, and N-ATF horses) at 5 minutes post-transport and returned to baseline by 30 minutes post-transport. Predominant parasympathetic nervous system (PNS) activity was observed during the first few hours and returned to baseline until the destination was reached. A recurrent, increased PNS activity was detected post-transport. Interaction effects of air condition-by-loading condition-by-time, air condition-by-time, and separate effects of air condition and time were observed on HR and various HRV variables during transport. A transient increase in beat-to-beat intervals, coinciding with decreased HR, was observed in ATF horses. The PNS index increased, corresponding to a decreased sympathetic nervous system index, in ATS horses during transport. We suggest that medium-distance road transport causes no stress for transport-experienced horses in a tropical environment. Air and loading conditions impacted hormonal and autonomic modulation, causing different responses in horses transported in differently conditioned trucks.

**Data Availability Statement:** HRV raw data and supplementary files are available at https://www.doi.org/10.6084/m9.figshare.25012616.

**Funding:** The author(s) received no specific funding for this work.

**Competing interests:** The authors have declared that no competing interests exist.

## Introduction

Equestrian sports are growing in popularity worldwide [1]. According to the Fédération Equestre Internationale (FEI) database (https://data.fei.org/Default.aspx), approximately 3,200 international shows are organized annually. Moreover, 478,000 horses are entered into the FEI database, of which nearly 77,000 are registered annually by the National Federations. An increasing number of horses participating in equestrian shows parallels with a rapid increase in horse transport by air [2, 3] and road [4, 5]. Of particular relevance to this study is that horses are almost always transported to the competition venue by road for at least some of the journey.

It is well-recognized that road transport is one of the stressful conditions that athletic horses encounter during the competition season [6, 7], in addition to training and competition [8, 9]. Stress responses during road transport have been evaluated by the determination of cortisol levels [10, 11] or cortisol levels in conjunction with heart rate variability [4, 5, 12]. Cortisol levels, a valid stress parameter related to the hypothalamic-pituitary-adrenocortical axis [13], are released following stressful challenges [14]. Heart rate variability (HRV), a rhythmic fluctuation in time intervals between heartbeats under the influence of sympathetic and parasympathetic (vagal) components, is also used to represent the sympatho-adrenomedullary axis associated with the state of the autonomic nervous system and in turn, stress responses in horses [15, 16]. An increase in cortisol levels, coinciding with decreased beat-to-beat (RR) intervals and the root mean square of successive RR interval differences (RMSSD), has been observed in horses during road transport, indicating a reduced vagal tone [4, 5].

The humid tropics, characterized by persistent high humidity and temperature, are associated with pronounced impacts on animals' physiological responses [17] and a high incidence of animal diseases [18]. Since humidity and temperature affect heat dissipation, an increase in both humidity and temperature decreases the rate of heat loss in animals [19, 20]. The heat stress and disease challenges can be exacerbated by poor housing conditions [21, 22] and nutritional management [23]. Since road transport has been considered a stressor and involves adverse effects regarding injuries and gastrointestinal disease [24–26], it may be of more significant concern during transport in humid tropics.

To date, the stress response in horses during road transport has primarily been reported in temperate environments. Unfortunately, no reports have been documented in tropical environments, particularly in different truck conditions. Therefore, the present study aimed to investigate the impact of medium-distance road transport with differently conditioned trucks (i.e., air-conditioned or non-air-conditioned) in a tropical environment. It was hypothesized that horses would demonstrate distinct stress responses under these different conditions.

## Materials and methods

### Horses

Six healthy athletic horses (two geldings and four mares, aged 11.83 ± 2.64 years and weighing 466.30 ± 38.70 kg) from the Horse Lover's Club, Pathum Thani, Thailand (co-ordinates 13.99432, 100.68082), were enrolled in this study. All the horses had previous experience in various national and international jumping events and were familiar with transportation by trucks and trailers. They were individually housed in 4 x 5 m2 stables and were fed 1 kg of commercial pellets three times a day (totaling 3 kg/day). Hay and tap water were available to them at all times. They trained for jumping competitions three to four days per week, with Mondays off. The horses did not undergo any medical or surgical treatments at the beginning of the study. The authors obtained verbal consent from the owners to include their horses in

the study. This research was approved by the Ethics Committee of the Institutional Animal Care and Use Committee, Kasetsart University (ACKU63-VET-020).

## Experimental protocol

Horses were assigned for repeated transport by four different truck conditions, including an air-conditioned truck with a full load (ATF) on the first day and a non-air-conditioned truck with a full load (N-ATF) on the second day. On the third day of the experiment, half the horses were transported in an air-conditioned truck with space load (ATS-D3) simultaneously with the other half in a non-air-conditioned truck with space load (N-ATS-D3); each horse experienced the other condition on day 4, in a cross-over design (Table 1). Road transports were arranged on the non-training day (Mondays, i.e., at seven-day intervals) over four consecutive weeks.

Before the transport, trucks were sanitized with an iodine solution spray. Hay nets were hung within the truck's accommodating room (Fig 1A). Horses were loaded in every space in the 'full' condition (or in alternate spaces in the 'space' condition), perpendicular to the truck's long axis, in a similar sequence in all trips. They were tied loosely with a cord approximately 50 cm long in front of the horse. They were separated by transparent partitions, permitting visual contact (Fig 1B–1D). Air and non-air-conditioned trucks were driven by familiar drivers in all transports, which were arranged as a non-stop round-trip on the highway heading to the venues where events are usually organized, returning by the same road to the origin. Hay was provided (but no water or pellets) throughout the transport.

## Data collection

**Driving speed, covered distance, and the trucks' internal environment during road transport.** Driving speed (km/h) and covered distances (km) were exported from a Polar sports watch (Vantage 2; Polar Electro Oy, Kempele, Finland), which was put within each truck, primarily for recording horses' RR intervals during transport. Relative humidity (%) and ambient temperature (°C) within the trucks were measured at 15-minute intervals using a temperature and humidity data logger (TM-305U; Tenmars Electronics, Taipei, Taiwan).

**Blood cortisol determination.** Blood cortisol levels were determined at before transport and 5, 30, and 90 minutes post-transport. At each time point, 5 ml of blood was withdrawn from the jugular vein and placed in blood collection tubes without anticoagulants. Serum samples were separated and preserved at –80°C for blood cortisol analysis, which was determined using a competitive chemiluminescent enzyme immunoassay (IMMULITE Analyzers, Siemens Healthineers, Erlangen, Germany) and expressed as nmol/L

**Table 1. Trucks' driving speed, distances covered, and onboard environment during road transport on four experiment dates.**

| Experiment dates | Transport conditions | Driving speed (km/h) | Distance (km) | Transport time (min) |
|---|---|---|---|---|
| Day 1 | ATF | 56.09 | 265.26 | 265 |
| Day 2 | N-ATF | 57.89 | 269.70 | 260 |
| Day 3 | ATS | 64.40 | 265.14 | 248 |
| | N-ATS | 64.15 | 265.26 | 249 |
| Day 4 | ATS | 65.17 | 265.08 | 245 |
| | N-ATS | 65.12 | 265.17 | 246 |
| Mean ± SD | | 62.14 ± 4.05 | 265.90 ± 1.84 | 252.20 ± 8.28 |

**ATF**, air-conditioned truck with a full load; **ATS**, air-conditioned truck with a space load; **N-ATF**, non-air-conditioned truck with a full load; **N-ATS**, non-air-conditioned truck with a space load.

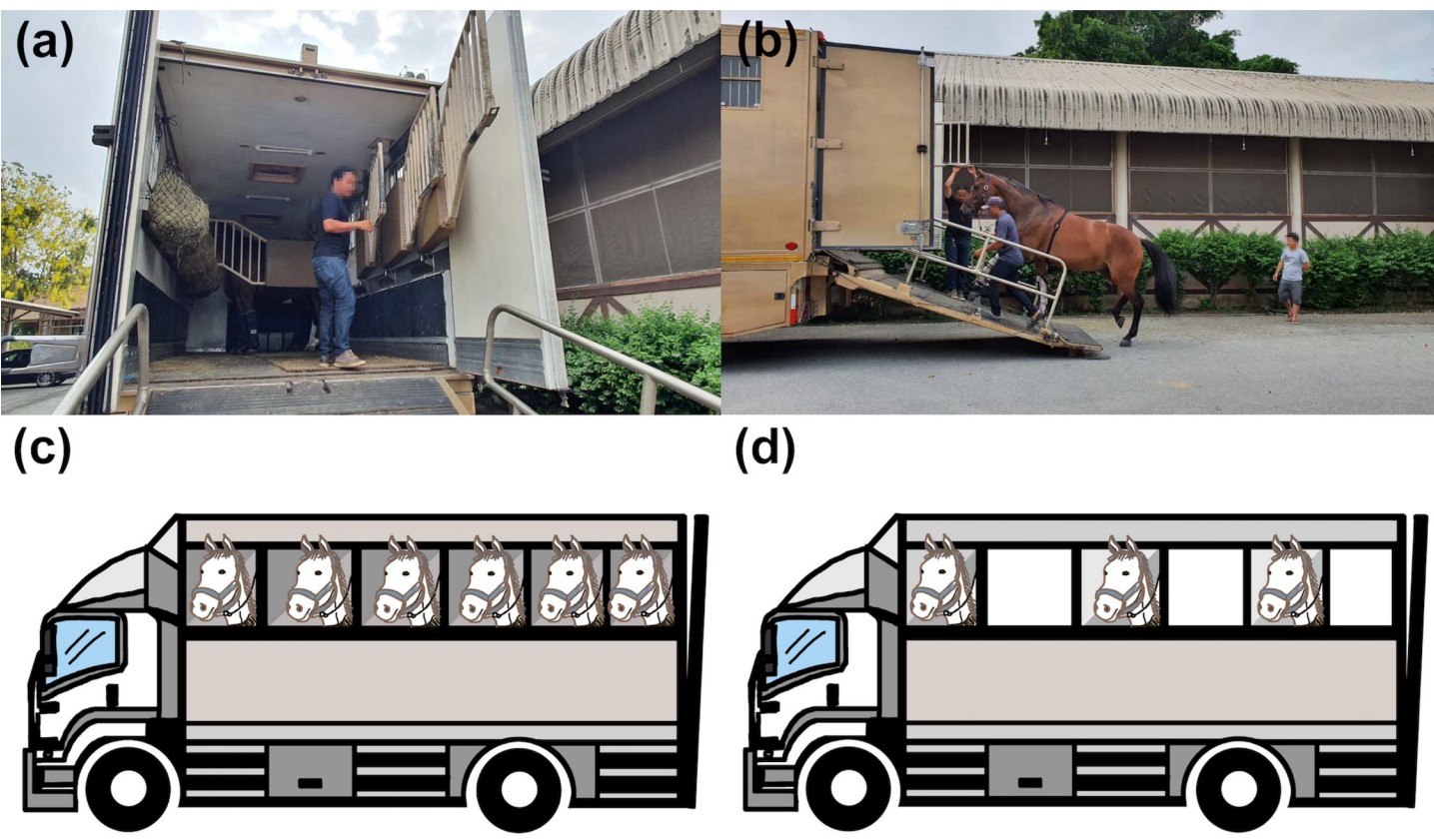

**Fig 1. Truck preparation and loading condition before transport.** Horses are separated by metal partitions, allowing them visual contact. (a) Hay nets are hung within the truck throughout transport. (b) A horse equipped with a Polar heart rate monitoring device is loaded onto the truck, in either (c) a full load of six horses or (d) a space load of three horses. Original copyright [2024].

**Heart rate and heart rate variability.** Horses were equipped with heart rate monitoring (HRM) devices to record RR intervals throughout the study period. In brief, the Polar equine belt, attached to an HR sensor (H10; Polar Electro Oy, Kempele, Finland), was soaked in water to increase connectivity and the transmission signal. Ultrasound gel was also applied to the belt before fastening around the horse's chest, and the HR sensor pocket was placed in the middle of the chest's left side. The device sets were installed on horses 30 minutes before transport to familiarize them with the device. Finally, the sensor was connected to the Polar sports watch to record RR intervals 15 minutes before transport, and this recording continued until 90 minutes post-transport. Sports watches were kept securely within the truck during the transport.

RR interval data from the Polar sports watch were uploaded to the Polar flow program (available at https://flow.polar.com/), and then HRV variables were computed by the Kubios HRV Scientific program (available at https://www.kubios.com/hrv-scientific/). Misaligned and ectopic beats were corrected by the software program's automatic artifact correction algorithm. Automatic noise detection was set at medium level. The smoothness priors method was set at 500 ms to remove RR time series non-stationarities. The HRV variables, exported from the Kubios program, were as follows:

1. *Time domain analysis*: HR, RR intervals, the standard deviation of normal-to-normal RR intervals (SDNN), the root mean square of successive RR interval differences (RMSSD), the relative number of successive RR interval pairs that differ by more than 50 msec (pNN50), and the stress index.

2. *Frequency domain analysis*: very-low-frequency (VLF) band (by default 0–0.01 Hz), low-frequency (LF) band (by default 0.01–0.07 Hz), high-frequency (HF) band (by default 0.07–0.6 Hz), total power, LF/HF ratio, and respiratory rate (RESP).

3. *Nonlinear analysis*: the standard deviation in the Poincaré plot perpendicular to the line-of-identity (SD1) and along the line-of-identity (SD2), as well as the SD2/SD1 ratio.

4. *Autonomic nervous system index*: parasympathetic nervous system (PNS) and sympathetic nervous system (SNS) indices.

The HR and HRV variables were estimated 15 minutes pre-transport (control) and at 15-minute intervals during transport until 90 minutes post-transport.

## Data analysis

Data were statistically analyzed using GraphPad Prism version 10.2.1 (GraphPad Software Inc, San Diego, USA). The Greenhouse–Geisser correction was automatically selected to estimate an epsilon (sphericity) and correct for lack of sphericity before analyses. Three-way repeated measures ANOVA was used to evaluate the interactions and main effects of air condition, loading condition, and time on changes in cortisol levels, HR, and HRV variables following road transports with differently conditioned trucks. Tukey's *post hoc* test was later applied to assess differences within and between groups at given time points. In the case of the one-grouping variable analysis, the Shapiro-Wilk test was used to verify the normal distribution of the data. As the data were normally distributed, an ordinary one-way ANOVA followed by Tukey's *post hoc* test was applied to determine the difference in internal humidity between trucks with different transport conditions. The Kruskal-Wallis test and Dunn's post hoc tests were employed to assess the internal temperatures between trucks with different transport conditions because the data showed a non-normal distribution. Correlations between trucks' internal environment and HRV variables were computed using the Spearman rank correlation ($r_s$) method because the data were non-normally distributed. The strength of correlation coefficients was reported as weak ($0.1 \leq r_s < 0.4$), moderate ($0.4 \leq r_s < 0.7$), and strong ($r_s \geq 0.7$) [27]. Eta squared ($\eta^2$) was used to estimate the effect size, as very small ($\eta^2 < 0.01$), small ($0.01 \leq \eta^2 < 0.06$), medium ($0.06 \leq \eta^2 < 0.14$), and large ($\eta^2 \geq 0.14$) [28]. When necessary, r-square ($r^2$) was alternatively applied to evaluate the effect size as small ($0.02 \leq r^2 < 0.13$), medium ($0.13 \leq r^2 < 0.26$), and large ($r^2 \geq 0.26$) [28]. Data were expressed as means ± *SD*. Statistical significance was set at $p < 0.05$.

## Results

### Driving speed, covered distance, and the trucks' internal environment during road transport

The truck's driving speed, distance covered, and internal environment during road transport are shown in Table 1. Trucks were driven at the average speed of 62.14 ± 4.05 km/h, covered 265.90 ± 1.84 km, and completed the transport trip within 252.20 ± 8.28 minutes. Relative humidity in ATF, N-ATF, ATS-D3, N-ATS-D3, ATS-D4 and N-ATS-D4 was 78.04 ± 4.38%, 55.13 ± 6.86%, 67.72 ± 5.58%, 56.26 ± 6.04%, 68.78 ± 5.58%, and 50.67 ± 4.91%, respectively. The relative humidity was higher in air-conditioned trucks than in non-air-conditioned trucks during transport (ATF vs. N-ATF, ATS-D3 vs. N-ATS-D3, ATS-D4 vs. N-ATS-D4;

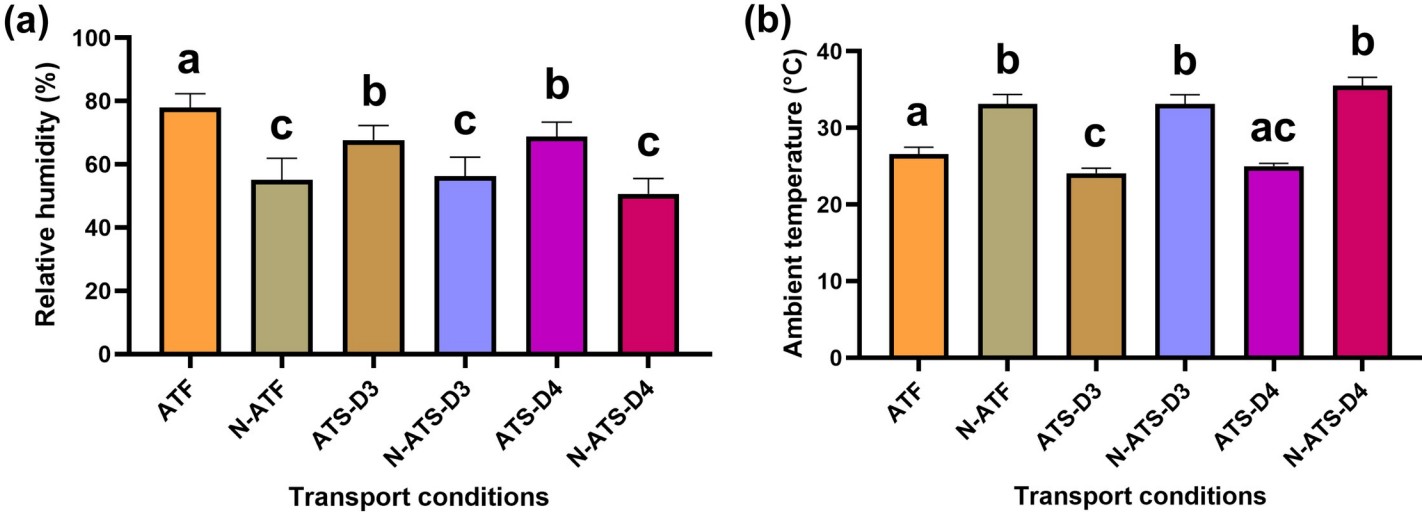

**Fig 2.** Relative humidity (a) and ambient temperature (b) within distinctly conditioned trucks. Different letters (a, b, and c) indicate a statistical difference between pairs of comparisons. **ATF**, air-conditioned truck with a full load; **N-ATF**, non-air-conditioned truck with a full load, **ATS-D3**, air-conditioned truck with a space load on day 3; **N-ATS-D3**, non-air-conditioned truck with a space load on day 3; **ATS-D4**, air-conditioned truck with a space load on day 4; **N-ATS-D4**, non-air-conditioned truck with a space load on day 4.

$F (5, 103) = 72.70, p < 0.0001, r^2 = 0.779$ for all pairs of comparison). Moreover, the relative humidity in ATF was higher than other transport conditions (ATF vs. N-ATF, ATS-D3, N-ATS-D3, ATS-D4, and N-ATS-D4; $F (5, 103) = 72.70, p < 0.0001, r^2 = 0.779$ for all pairs of comparison) (Fig 2A). Ambient temperature in ATF, N-ATF, ATS-D3, N-ATS-D3, ATS-D4, and N-ATS-D4 were 26.58 ± 0.90˚C, 33.17 ± 1.21˚C, 24.09 ± 0.68˚C, 33.11 ± 1.25˚C, 25.01 ± 0.37˚C, and 35.50 ± 1.11˚C, respectively. The temperature was not different when compared among non-air-conditioned trucks. Although the temperature in ATS-D4 did not differ from ATF and ATS-D3 conditions, the temperature in ATS-D3 was lower than in the ATF condition ($z = 2.95, p = 0.048, \eta^2 = 0.891$). The temperature was lower in air-conditioned trucks than in non-air-conditioned trucks (ATF vs. N-ATF: $z = 3.23, p = 0.0186, \eta^2 = 0.891$; ATS-D3 vs. N-ATS-D3: $z = 5.78, p < 0.0001, \eta^2 = 0.891$; ATS-D4 vs. N-ATS-D4: $z = 6.68, p < 0.0001, \eta^2 = 0.891$) (Fig 2B).

### Blood cortisol levels

An air condition x loading condition x time 3-way interaction ($F (3, 60) = 3.49, p = 0.021, \eta^2 = 0.022$) and an independent effect of time ($F (2.17, 43.40) = 83.77, p < 0.0001, \eta^2 = 0.523$) was observed on blood cortisol levels (Table 2). Compared to their baselines, horses showed a non-

**Table 2. Blood cortisol levels (mean ± SD) in horses in differently conditioned trucks during road transport.**

| Periods | Blood cortisol levels (nmol/L) | | | |
|---|---|---|---|---|
| | **ATF** | **ATS** | **N-ATF** | **N-ATS** |
| **Pre-transport** | 69.46 ± 32.25 | 84.64 ± 17.43 | 83.72 ± 19.18 | 66.24 ± 11.97 |
| **5 min post-transport** | 127.88 ± 32.25 | 109.94 ± 29.49 | 120.52 ± 25.34 | 117.76 ± 25.46* |
| **30 min post-transport** | 83.26 ± 25.97 | 73.6 ± 19.88 | 86.48 ± 21.92 | 84.18 ± 23.60 |
| **90 min post-transport** | 61.18 ± 17.41 | 51.98 ± 17.14 | 59.8 ± 14.57 | 54.28 ± 14.25 |

* indicates a significant difference when compared to pre-transport. **ATF**, air-conditioned truck with a full load; **ATS**, air-conditioned truck with a space load; **N-ATF**, non-air-conditioned truck with a full load; **N-ATS**, non-air-conditioned truck with a space load.

significant increase in blood cortisol levels 5 minutes post-transport with ATF ($p = 0.051$, $df = 5$), N-ATF ($p = 0.122$, $df = 5$), and ATS ($p = 0.600$, $df = 5$) conditions. However, cortisol levels increased significantly in N-ATS horses 5 minutes post-transport ($p = 0.022$, $df = 5$). However, cortisol levels returned to the baseline values within 30 and 90 minutes in all transport conditions (ATF, 30 minutes: $p = 0.992$, $df = 5$; 90 minutes: $p = 0.999$, $df = 5$) (N-ATF, 30 minutes: $p = 0.999$, $df = 5$; 90 minutes: $p = 0.250$, $df = 5$) (ATS, 30 minutes: $p = 0.968$, $df = 5$; 90 minutes: $p = 0.390$, $df = 5$) (N-ATS, 30 minutes: $p = 0.650$, $df = 5$; 90 minutes: $P = 0.590$, $df = 5$) (Table 2).

## Heart rate and heart rate variability

**HR, RR intervals, and ANS index.** The air condition x time interaction and separate main effects of air condition and time were detected on the modulation in both RR intervals (interaction: $F(22, 440) = 2.72$, $p < 0.0001$, $\eta^2 = 0.037$; air condition: $F(1, 20) = 5.30$, $p = 0.032$, $\eta^2 = 0.100$; time: $F(2.48, 49.53) = 13.36$, $p < 0.0001$, $\eta^2 = 0.181$) and HR (interaction: $F(22, 440) = 2.82$, $p < 0.0001$, $\eta^2 = 0.036$; air condition: $F(1, 20) = 4.92$, $p = 0.049$, $\eta^2 = 0.095$; time: $F(3.09, 61.71) = 15.02$, $p < 0.0001$, $\eta^2 = 0.194$). There were significant effects of the air condition x time interaction ($F(22, 440) = 1.59$, $p = 0.046$, $\eta^2 = 0.016$) and time (F $(2.35, 46.95) = 6.73$, $p = 0.002$, $\eta^2 = 0.069$) on the PNS index. On the other hand, the air condition x loading condition x time interaction ($F(22, 440) = 1.59$, $p = 0.044$, $\eta^2 = 0.018$), air condition x time interaction ($F(22, 440) = 2.23$, $p = 0.001$, $\eta^2 = 0.025$), and the independent effect of time ($F(3.35, 66.90) = 12.65$, $p < 0.0001$, $\eta^2 = 0.14$) on SNS index were all statistically significant (Fig 3).

Although the RR intervals and HR at the end of transport (after 240 minutes) did not differ from the pre-transport value, fluctuations in these variables were observed during transport. There was a transient increase in RR intervals at 180 minutes compared to 90 ($p = 0.032$, $df = 5$), 120 ($p = 0.040$, $df = 5$), and 135 minutes ($p = 0.038$, $df = 5$) in ATF horses; however, the increase was not quite significant when compared to the control ($p = 0.055$, $df = 5$). The increase was also detected at 180 minutes compared to 150 minutes in N-ATS horses ($p = 0.040$, $df = 5$). Compared to the control, RR intervals rose at 60 minutes post-transport in N-ATF horses (Fig 3A). HR was unchanged during transport in N-ATF, ATS, and N-ATS horses. Nonetheless, ATF horses had a reduced HR at 180 minutes compared to 90 minutes during the transport ($p = 0.005$, $df = 5$) (Fig 3B).

In ATS horses, the PNS index increased progressively at 105 ($p = 0.022$, $df = 5$), 150 ($p = 0.015$, $df = 5$), 165 ($p = 0.008$, $df = 5$), 210 ($p = 0.040$, $df = 5$), and 225 minutes ($p = 0.030$, $df = 5$), until the end of transport at 240 minutes ($p = 0.041$, $df = 5$). The PNS index returned to its control value 90 minutes post-transport. In N-ATS horses, the PNS index was non-significantly reduced at 15 minutes during transport ($p = 0.774$, $df = 5$). It then increased at 60 ($p = 0.018$, $df = 5$), 75 ($p = 0.009$, $df = 5$), and 90 minutes ($p = 0.001$, $df = 5$) to reach the control value at 105 minutes during transport. Nevertheless, there was no difference in PNS index in ATF, N-ATF, and N-ATS horses at the end of transport (after 240 minutes) or during the 90 minutes post-transport (Fig 3C). SNS index was lower at 165 minutes compared to 75 minutes in N-ATF horses ($p = 0.022$, $df = 5$) and at 225 compared to 45 minutes in ATS horses ($p = 0.045$, $df = 5$) during transport. Even though the SNS index at the end of transport did not differ from the control in ATF, N-ATF, and N-ATS horses, a reduced SNS index was observed at the end of transport, albeit a non-significant difference ($p = 0.096$, $df = 5$). In N-ATS horses, the SNS index was lower at 75 minutes post-transport compared to 135 minutes during transport ($p = 0.048$, $df = 5$) and at 90 minutes post-transport compared to 90 ($p = 0.045$, $df = 5$) and 135 minutes ($p = 0.032$, $df = 5$) during transport (Fig 3D).

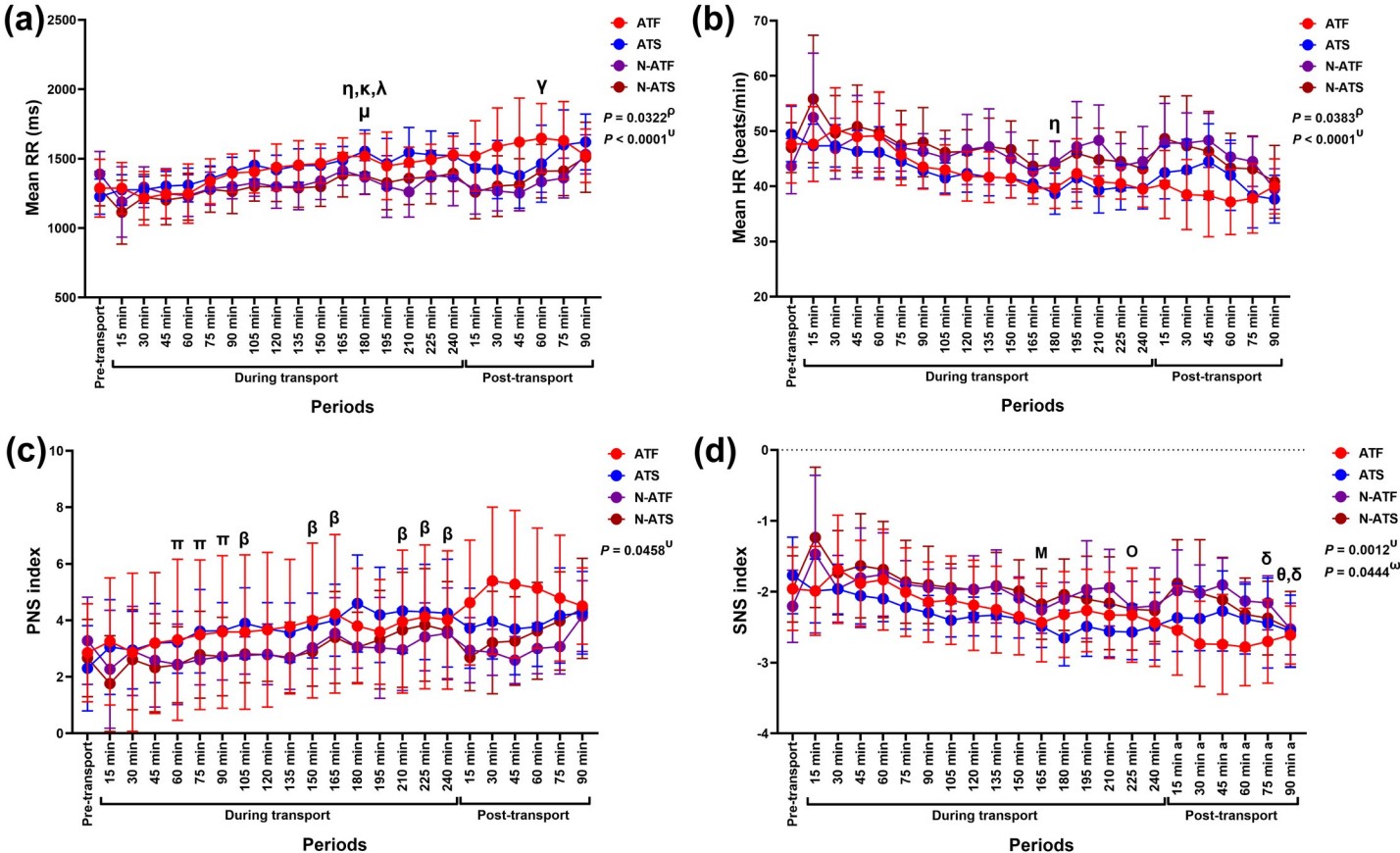

**Fig 3.** Modulation in (a) mean HR, (b) mean RR intervals, (c) PNS index, and (d) SNS index following road transport with different air and loading conditions. [ρ] indicates the independent effect of air condition on variable modification. [υ] indicates the effect of the air condition x time interaction on variable modification. [ω] indicates the effect of the air condition x loading condition x time interaction on variable modification. [γ] indicates a significant difference in N-ATF horses' variables compared to the pre-transport (control). [η, κ, λ] indicate significant differences in ATF horses' variables at 90, 120, and 135 minutes, compared to 180 minutes during transport. [μ] indicates a significant difference in N-ATS horses' variables between 150 and 180 minutes during transport. [β] indicates a significant difference in ATS horses' variables compared to the pre-transport (control). [π] indicates a significant difference in N-ATS horses' variables between given time points and 15 minutes during transport. [M] indicates a significant difference in N-ATF horses' variables between 75 and 165 minutes during transport. [O] indicates a significant difference in ATS horses' variables between 45 and 225 minutes during transport. [θ] indicates a significant difference in N-ATS horses' variables between a given time point and 90 minutes during transport. [δ] indicates a significant difference in N-ATS horses' variables between given time points and 135 minutes during transport. **ATF**, air-conditioned truck with a full load; **ATS**, air-conditioned truck with a space load; **N-ATF**, non-air-conditioned truck with a full load; **N-ATS**, non-air-conditioned truck with a space load; **HR**, heart rate; **RR**, beat-to-beat intervals.

**Time-domain results.** The air condition x loading condition x time interaction was observed on pNN50 ($F_{(22, 440)} = 1.60$, $p = 0.042$, $\eta^2 = 0.020$). Whereas the air condition x loading condition x time interaction and the main effect of time affected modification of stress index (interaction effect: $F_{(22, 440)} = 1.60$, $p = 0.043$, $\eta^2 = 0.031$; time effect: $F_{(4.97, 99.38)} = 6.49$, $p < 0.0001$, $\eta^2 = 0.124$). The main effect of time was observed on SDNN ($F_{(3.93, 78.61)} = 7.24$, $p < 0.0001$, $\eta^2 = 0.118$). However, there were no significant interactions or main effects on RMSSD (Fig 4).

Although the SDNN increased in horses at 15, 30, and 45 minutes post-transport, a significant difference was detected only at 30 minutes post-transport ($p = 0.003$, $df = 23$), while near-significant differences were revealed at 15 ($P = 0.065$, $df = 23$) and 45 minutes ($p = 0.081$, $df = 23$) post-transport (Fig 4A). RMSSD reduced in horses at 90 minutes post-transport ($p = 0.040$, $df = 23$) (Fig 4B). Despite showing the interaction effect, no change was observed

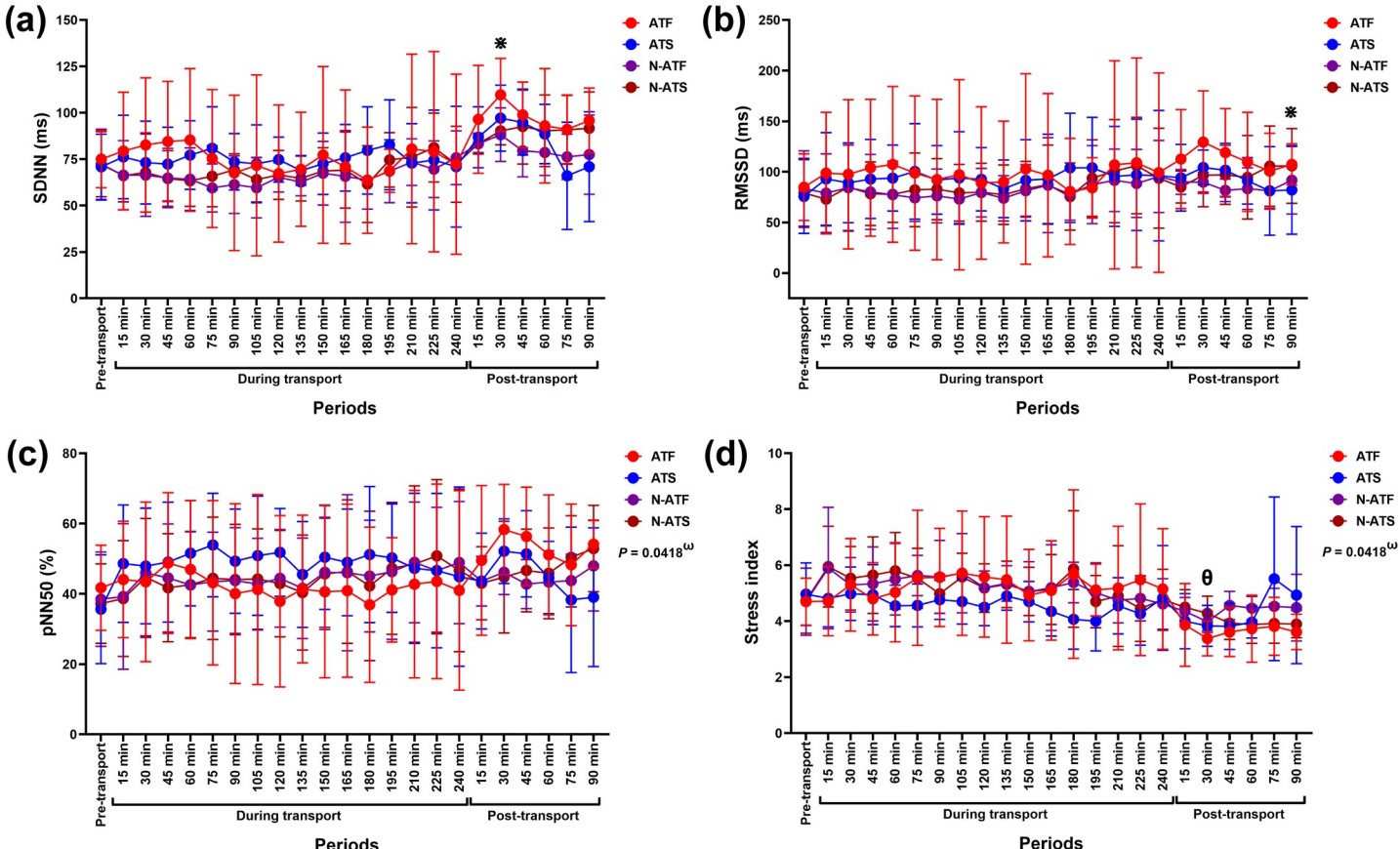

**Fig 4.** Modulation in (a) SDNN, (b) RMSSD, (c) pNN50, and (d) stress index following road transport with different air-conditioning and loading conditions. $^{\omega}$ indicates the effect of the air-condition x loading condition x time interaction on variable modification. $^{*}$ indicates a significant difference between variables of all horses at given time points and pre-transport (control). $^{\theta}$ indicates a significant difference in N-ATS horses' variables between a given time point and 90 minutes during transport. **ATF**, air-conditioned truck with a full load; **ATS**, air-conditioned truck with a space load; **N-ATF**, non-air-conditioned truck with a full load; **N-ATS**, non-air-conditioned truck with a space load; **SDNN**, the standard deviation of the normal-to-normal beat interval; **RMSSD**, the root mean square of successive RR interval differences; **RR**, beat-to-beat intervals; **pNN50**, the relative number of the beat-to-beat intervals that differs from 50 ms.

on pNN50 in ATF, N-ATF, ATS, and N-ATS horses during or after transport (Fig 4C). The stress index did not change in ATF, N-ATF, and ATS horses during and after transport. It was lower at 30 minutes post-transport compared to 90 minutes in N-ATS horses during the transport ($p = 0.017$, $df = 5$) (Fig 4D).

## Frequency-domain results

Time affected modulation in total power ($F (3.96, 79.21) = 6.61$, $p = 0.0001$, $\eta^2 = 0.123$), %VLF ($F (6.12, 122.3) = 7.98$, $p < 0.0001$, $\eta^2 = 0.218$), %LF ($F (5.20, 104.0) = 20.13$, $p < 0.0001$, $\eta^2 = 0.239$), and %HF ($F (5.20, 104.0) = 20.13$, $p < 0.0001$, $\eta^2 = 0.239$). However, the air condition x loading condition x time interaction impacted RESP modification ($F (22, 440) = 2.86$, $p < 0.0001$, $\eta^2 = 0.065$) (Fig 5). There was a main effect of time on LF/HF ratio ($F (5.78, 115.5) = 13.31$, $p < 0.0001$, $\eta^2 = 0.214$).

Total power band increased at 30 minutes post-transport ($p = 0.009$, $df = 23$), while %VLF decreased over time ($p < 0.05$–$0.001$, $df = 23$) until returning to the control value at 75–90 minutes post-transport (Fig 5A and 5B). The %LF decreased transiently, corresponding to an

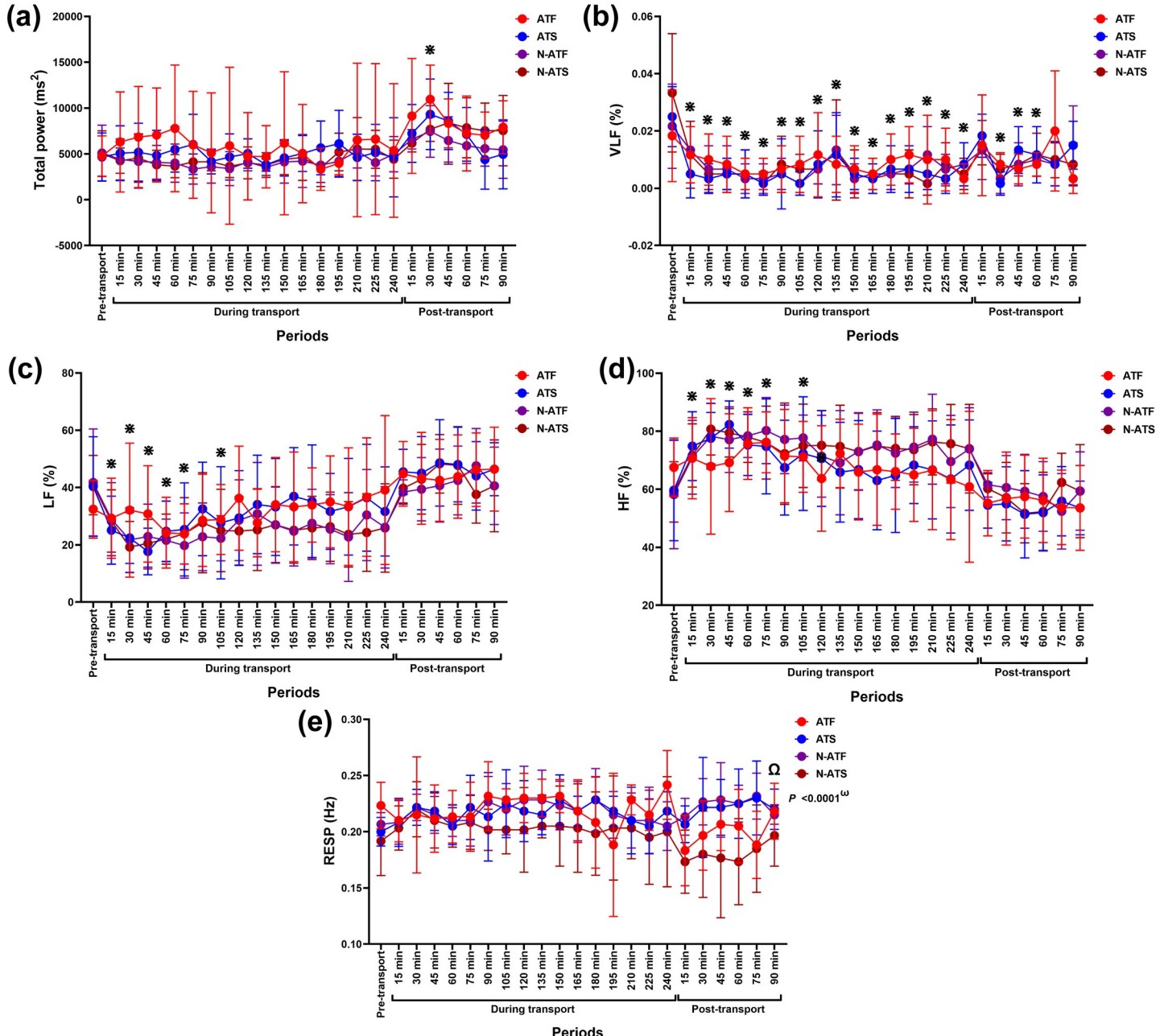

**Fig 5.** Total power (a), %VLF (b), %LF (c), %HF (d) contributions, and RESP modification (e) following road transport with different air and loading conditions. **ω** indicates the effect of the air condition x loading condition x time interaction on variable modification. **\*** indicates a significant difference between variables of all horses at given time points and pre-transport (control). **Ω** indicates a significant difference in ATF horses' variables between 15 and 90 minutes post-transport. **ATF**, air-conditioned truck with a full load; **ATS**, air-conditioned truck with a space load; **N-ATF**, non-air-conditioned truck with a full load; **N-ATS**, non-air-conditioned truck with a space load; **VLF**, very-low-frequency band; **LF**, low-frequency band; **HF**, high-frequency band.

increase in the %HF at 15–105 minutes during transport ($p < 0.05$–0.001, $df = 23$ for both variables). The variables returned to the control value at 120 minutes during transport afterward (Fig 5C and 5D). In ATF horses, RESP was higher at 90 minutes post-transport than at 15 minutes post-transport ($p = 0.013$, $df = 5$). The RESP in N-ATF, ATS, and N-ATS horses did not change across the study period (Fig 5E). A short-term reduction in LF/HF ratio was observed

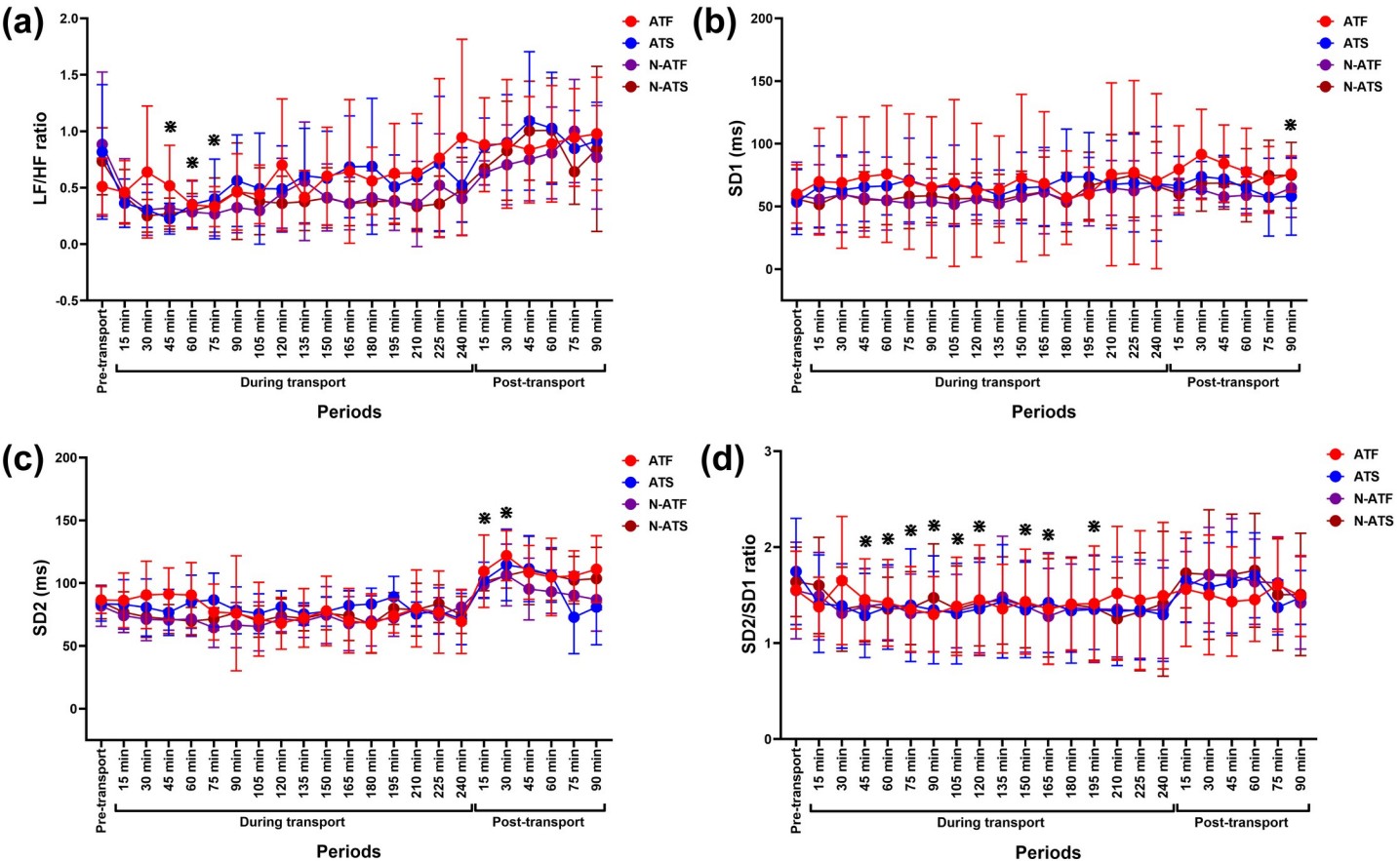

**Fig 6.** LF/HF ratio (a), SD1 (b), SD2 (c), and SD2/SD1 ratio (d) following road transport with different air and loading conditions. * indicates a significant difference between variables of all horses at given time points and pre-transport (control). **ATF**, air-conditioned truck with a full load; **ATS**, air-conditioned truck with a space load; **N-ATF**, non-air-conditioned truck with a full load; **N-ATS**, non-air-conditioned truck with a space load; **LF**, low-frequency band; **HF**, high-frequency band; **SD1**, the standard deviation of Poincaré plot perpendicular to the line-of-identity; **SD2**, the standard deviation of Poincaré plot along the line-of-identity.

at 45–75 minutes during transport ($p < 0.05$, $df = 23$ for all time points) before rising to the control value at 90 minutes during transports afterward (Fig 6A).

## Nonlinear results

There were no main or interaction effects on the SD1 variable. However, time affected changes in SD2 ($F$ (5.44, 108.9) = 14.16, $p < 0.0001$, $\eta^2 = 0.282$) and SD2/SD1 ratio ($F$ (3.77, 75.47) = 4.73, $p = 0.002$, $\eta^2 = 0.047$) (Fig 6).

The SD1 increased at 90 minutes post-transport ($p = 0.039$, $df = 23$) (Fig 6B), whereas SD2 rose at 15 ($p = 0.016$, $df = 23$) and 30 minutes ($p = 0.003$, $df = 23$) post-transport (Fig 6C). The SD2/SD1 ratio decreased at 45–195 minutes during transport ($p < 0.05$, $df = 23$ for all time points) (Fig 6D).

## Correlations between trucks' internal environment and HRV variables during road transport

The correlation among HRV variables and relative humidity (RH) and temperature within the trucks are shown in Table 3 and the correlation matrices in the supplementary files (S1–S4

**Table 3. Correlation between internal humidity and temperature and HRV variables in ATF, ATS, N-ATF, and N-ATF transport conditions.**

| Variables | Spearman correlation coefficients ($r_s$) | | | | | | | |
|---|---|---|---|---|---|---|---|---|
| | ATF | | ATS | | N-ATF | | N-ATS | |
| | RH (%) | Temp (°C) | RH (%) | Temp (°C) | RH (%) | Temp (°C) | RH (%) | Temp (°C) |
| Mean RR | −0.626 | 0.566 | −0.961 | −0.743 | – | – | – | – |
| Mean HR | 0.580 | −0.543 | 0.972 | 0.724 | – | – | – | – |
| SDNN | – | – | – | – | – | 0.502 | −0.844 | 0.544 |
| RMSSD | – | – | −0.534 | – | – | 0.502 | −0.620 | – |
| %LF | −0.470 | 0.483 | −0.475 | −0.725 | −0.794 | 0.572 | – | – |
| %HF | 0.470 | −0.483 | 0.475 | 0.725 | 0.794 | −0.572 | – | – |
| LF/HF ratio | – | 0.449 | −0.505 | −0.725 | −0.803 | 0.602 | – | 0.490 |
| SD1 | – | – | −0.561 | – | −0.430 | 0.502 | −0.630 | – |
| SD2 | – | – | – | – | −0.684 | 0.593 | −0.744 | 0.542 |
| SD2/SD1 ratio | – | – | 0.525 | – | −0.650 | 0.489 | – | – |
| PNS index | −0.631 | 0.544 | −0.951 | −0.725 | – | – | −0.536 | – |
| SNS index | 0.599 | −0.531 | 0.929 | 0.741 | – | – | 0.605 | – |
| Stress index | – | – | 0.563 | 0.482 | 0.665 | −0.700 | 0.816 | −0.519 |

ATF, air-conditioned truck with a full load; ATS, air-conditioned truck with a space load; N-ATF, non-air-conditioned truck with a full load; N-ATS, non-air-conditioned truck with a space load; RR, beat-to-beat intervals; HR, heart rate; SDNN, standard deviation of RR intervals; RMSSD, root mean square of successive RR differences; LF, low-frequency band; HF, high-frequency band; SD1, the standard deviation of Poincaré plot perpendicular to the line-of-identity; SD2, the standard deviation of Poincaré plot along the line-of-identity; PNS, parasympathetic nervous system; SNS, sympathetic nervous system.

Figs). Internal humidity was strongly and negatively correlated with internal temperature in ATF ($r_s = −0.754$, $p < 0.001$), N-ATF ($r_s = −0.727$, $p < 0.001$), and N-ATS conditions ($r_s = −0.826$, $p < 0.001$). In contrast, a positive correlation was detected between internal humidity and temperature in the ATS condition ($r_s = 0.721$, $p = 0.002$) (S1–S4 Figs).

RR intervals and HR were moderately correlated with internal humidity (RR intervals: $r_s = −0.626$, $p = 0.002$; HR: $r_s = 0.580$, $p = 0.006$) and temperature (RR intervals: $r_s = 0.566$, $p = 0.007$; HR: $r_s = −0.543$, $p = 0.011$) in ATF horses. In contrast, they strongly correlated with internal humidity (RR intervals: $r_s = −0.961$, $p < 0.001$; HR: $r_s = 0.972$, $p < 0.001$) and temperature (RR intervals: $r_s = −0.743$, $p = 0.001$; HR: $r_s = 0.724$, $p = 0.001$) in ATS horses. Nonetheless, RR intervals and HR changes did not correlate with the internal environment in N-ATF and N-ATS horses.

SDNN was not correlated with internal humidity or temperature in ATF and ATS horses; however, it showed a moderate correlation with internal temperature in N-ATF horses ($r_s = 0.502$, $p = 0.024$). It was also strongly correlated with internal humidity ($r_s = −0.844$, $p < 0.001$) and moderately with internal temperature ($r_s = 0.544$, $p = 0.026$) in N-ATS horses. RMSSD was not correlated with internal humidity or temperature in ATF horses, but was moderately correlated with internal humidity in ATS ($r_s = −0.534$, $p = 0.029$) and N-ATS horses ($r_s = −0.620$, $p = 0.009$). The RMSSD was moderately correlated with internal temperature in N-ATF horses ($r_s = 0.502$, $p = 0.024$).

In ATF horses, %LF and %HF were moderately correlated with internal humidity (%LF: $r_s = −0.470$, $p = 0.032$; %HF: $r_s = 0.470$, $p = 0.032$) and temperature (%LF: $r_s = 0.483$, $p = 0.026$; %HF: $r_s = −0.483$, $p = 0.026$). Notwithstanding this, the correlation was moderate with internal humidity (%LF: $r_s = −0.475$, $p = 0.048$; %HF: $r_s = 0.475$, $p = 0.048$) but strong with internal temperature (%LF: $r_s = −0.725$, $p = 0.001$; %HF: $r_s = 0.725$, $p = 0.001$) in ATS horses. On the other hand, %LF and %HF showed a strong correlation with internal humidity (%LF: $r_s = −0.794$, $p < 0.001$; %HF: $r_s = 0.794$, $p < 0.001$) but a moderate correlation with internal temperature

(%LF: $r_s$ = 0.572, $p$ = 0.008; %HF: $r_s$ = –0.572, $p$ = 0.008) in N-ATF horses. %LF and %HF modulation did not correlate with the internal environment in N-ATS horses. LF/HF ratio was moderately correlated with internal temperature in ATF ($r_s$ = 0.449, $p$ = 0.041) and N-ATS horses ($r_s$ = 0.490, $p$ = 0.048), while it had stronger correlations with humidity and temperature in ATS (RH: $r_s$ = –0.505, $p$ = 0.041; temperature: $r_s$ = –0.725, $p$ = 0.001) and N-ATF horses (RH: $r_s$ = –0.803, $p < 0.001$; temperature: $r_s$ = 0.602, $p$ = 0.005).

No correlations were observed between the internal environment and either SD1, SD2, or the SD2/SD1 ratio in ATF horses. In contrast, in N-ATF horses, correlations were detected involving internal humidity (SD1: $r_s$ = –0.430, $p$ = 0.048; SD2: $r_s$ = –0.684, $p$ = 0.001; SD2/SD1 ratio: $r_s$ = –0.650, $p$ = 0.002) and temperature (SD1: $r_s$ = 0.502, $p$ = 0.024; SD2: $r_s$ = 0.593, $p$ = 0.006; SD2/SD1 ratio: $r_s$ = 0.489, $p$ = 0.028). Internal humidity was correlated with SD1 ($r_s$ = –0.561, $p$ = 0.021) and SD2/SD1 ratio ($r_s$ = 0.525, $p$ = 0.033) in ATS horses and with SD1 ($r_s$ = –0.630, $p$ = 0.008) and SD2 ($r_s$ = –0.744, $p$ = 0.001) in N-ATS horses. The SD2 was also correlated with internal temperature ($r_s$ = 0.542, $p$ = 0.018) in N-ATS horses.

PNS and SNS indices were moderately correlated with internal humidity (PNS index: $r_s$ = –0.631, $p$ = 0.002; SNS index: $r_s$ = 0.599, $p$ = 0.004) and temperature (PNS index: $r_s$ = 0.544, $p$ = 0.011; SNS index: $r_s$ = –0.531, $p$ = 0.013) in ATF horses. However, they strongly correlated with internal humidity (PNS index: $r_s$ = –0.951, $p < 0.001$; SNS index: $r_s$ = 0.929, $p < 0.001$) and temperature (PNS index: $r_s$ = –0.725, $p$ = 0.001; SNS index: $r_s$ = 0.741, $p$ = 0.001) in ATS horses. No correlation was detected between PNS and SNS indices and internal environment in N-ATF horses; nonetheless, they correlated with internal humidity (PNS index: $r_s$ = –0.536, $p$ = 0.028; SNS index: $r_s$ = 0.605, $p$ = 0.012) in N-ATS horses.

There was no correlation between stress index and internal environment in ATF horses; however, these were correlated in ATS (RH: $r_s$ = 0.563, $p$ = 0.020; temperature: $r_s$ = 0.482, $p$ = 0.049), N-ATF (RH: $r_s$ = 0.665, $p$ = 0.001; temperature: $r_s$ = –0.700, $p$ = 0.001), and N-ATS horses (RH: $r_s$ = 0.816, $p < 0.001$; temperature: $r_s$ = –0.519, $p$ = 0.035).

## Discussion

The present study investigated blood cortisol levels, HR, and HRV variables in a tropical environment in response to road transport in differently conditioned trucks. The striking findings from the study were: 1) a three-way air condition x loading condition x time interaction effect was detected on modulations of blood cortisol levels, pNN50, stress index, RESP, and SNS index, while an air condition x time interaction effect was observed on RR intervals, HR, PNS index, and SNS index; 2) cortisol levels showed a non-significant increase in ATF, ATS, and N-ATF horses, and a significant increase in N-ATS horses, at 5-minutes post-transport; 3) SDNN, RMSSD, SD1, and SD2 did not change during the transport, but increased post-transport; 4) a transient increase in HF, coinciding with decreased LF, was observed during the first two hours of transport; 5) a short-term reduced LF/HF ratio, and longer-term reduced SD2/SD1 ratio, was also noticed during transport; 6) an increased RR interval, corresponding to decreased HR, was detected in ATF horses; and 7) PNS index rose, concurrent with a reduced SNS index, in ATS horses.

The results suggest that road transport caused no stress on horses, irrespective of the air-conditioning condition of trucks, during medium-distance transport in a tropical environment. However, the interaction of independent variables on hormonal and autonomic regulation led horses to respond distinctively to different transport conditions. An increase in HF, corresponding to decreased LF, LF/HF ratio, and SD2/SD1 ratio, indicated a shift toward vagal dominance and, in turn, transient relaxation in horses during transport. An increase in RR interval and reduced HR in ATF horses, simultaneously with a rise in PNS index and reduced

SNS index in ATS horses, may partly indicate temporal relaxation in transport-experienced horses within air-conditioned trucks during transport.

In this study, the internal humidity was higher in air-conditioned than in non-air-conditioned trucks. A difference in air-conditioning settings could explain this disparity. It has been reported that the relative humidity was significantly higher within a closed barn than in an outside barn [29, 30]. Likewise, the moisture production becomes pronounced, causing increased relative humidity in a closed-door, air-conditioned room of a human dwelling [31]. Accordingly, the higher humidity appeared to be due to elevated moisture production by the horses within the closed system of the air-conditioned trucks. More importantly, compared to the ATS condition, the greater humidity and temperature in the ATF condition coincided with an increased number of horses within the trucks. This result suggested that loading conditions also affected changes within the environment of air-conditioned transport vehicles.

An interaction between air condition, loading condition, and time was detected, leading to different cortisol modulation in horses with different transport conditions. Cortisol levels increased in all conditioned trucks soon after the end of transport; however, a significant difference was observed only in N-ATS horses. This finding may indicate, in part, that the N-ATS condition created more stress than the others. Cortisol levels are modulated by the circadian rhythm, peaking in the morning and then gradually reducing [29, 30, 32, 33]. Since our experiments were initiated in the morning and ended in the afternoon, it could be that increases in cortisol levels after transport were not under circadian rhythm but external stimuli from the transports. These results align with previous reports on the modulation of cortisol levels in horses after road transport [4, 5], even over short distances [9].

Although a significant effect of interaction was observed on cortisol modulations, it may not be adequate to verify the potential application of the results. In general, significant differences ($p$-values) among pairs of comparisons are frequently used in conjunction with the effect size (eta square: $\eta^2$ or r square: $r^2$) to estimate the practical significance of the research findings. A large effect size reflects the practical significance of the outcome, whereas the implication is limited if the effect size is small [34]. The present study detected a significant interaction effect but a small effect size, suggesting a narrow application of this finding. Thus, the fluctuation of cortisol levels in this study must be interpreted with caution. Although increases in cortisol levels did not reach statistical significance in the air-conditioned trucks (ATF and ATS horses), it was almost significant in the ATF horses at the end of the transport ($p = 0.051$). With somewhat similar internal temperature, a change in ATF horses' cortisol levels was supposed to correspond with the elevated humidity in the ATF condition during road transport. A significant rise in cortisol levels may have been reached following a further increased internal humidity if longer transport duration had been arranged.

Road transport causes a decrease in RR intervals, SDNN, and RMSSD variables in horses [4, 5, 12]. It also leads to a reduced SD1, corresponding to an increase in SD2 [4, 12] and HR [35, 36] in horses during transport. Since a decline in SDNN, RMSSD, and SD1 indicate reduced vagal tone [37, 38], it can confirm that road transport is stressful, causing a change in sympathovagal balance in horses. However, based on HR and HRV modulation, the present study contrasts with these previous studies, indicating no stress in horses during road transport. This notion was supported by a lack of changes in HRV variables, including SDNN, RMSSD, pNN50, stress index, SD1, and SD2 at given times during transport compared to pre-transport baselines. Nevertheless, the transient enhanced SDNN and SD2 at 30 minutes and RMSSD at 90 minutes post-transport were observed in all horses, corresponding to a reduced stress index at 30 minutes post-transport in N-ATS horses. Since SDNN and SD2 mirror long-term variation of the heart rate under the sympathetic and vagal influences [37, 39], the increased SDNN and SD2 at 30 minutes post-transport may reflect waxing and waning

sympathetic and vagal activities following free movement in the stables after an extended period of standing in the trucks. A later increase in RMSSD and SD1 indicated more relaxation 90 minutes after transport. Even though changes in these variables were most likely to result from voluntary movement after unloading, the effect of circadian rhythm on HRV modification cannot be ignored. It has been reported that vagal dominance was observed during the night, compared to daytime [15, 38, 40]. Since the transport began at approximately 09.00 h and ended back at the stables around 14.30–15.00 h, the effect of circadian rhythm on changes in time-domain and nonlinear variables after transports should be considered. Accordingly, these HRV modifications accompanying road transport should be interpreted carefully.

Modification of HR, RR intervals, and PNS and SNS indices differed in horses exposed to distinct transport conditions. Despite there being no differences compared to the pre-transport baseline in all groups, RR intervals demonstrated a transient increment in ATF and N-ATS horses during transport, while a decrease in HR was detected in ATF horses. Modulation of the ANS index also differed in horses with different transport conditions. An increase in the PNS index, corresponding to a decline in the SNS index, was observed, indicating increased vagal activity in ATS horses during transport. These results supported the notion that road transport caused no stress on transport-experienced horses. Indeed, since the air condition x time interaction was detected on changes in RR intervals, HR, PNS index, SNS index, and HRV variables in the ATF and ATS horses, it was assumed (at least in part) that the road transport with air-conditioned vehicles may tend to promote transient relaxation in horses during the road transport in tropical environments.

Since time-domain results merely quantify the net effect of the interplay between PNS and SNS, they are insufficient to indicate a distinct ANS activity and sympathovagal balance [37, 41, 42]. However, spectral analysis, which can distinguish the sympathetic and vagal influences, can be used to assess the sympathovagal balance [38]. Because the efferent vagal component primarily affects the maximal frequency of the HF band, the HF power band is an indicator of parasympathetic activity [39, 43], whereas the pharmacological blockade of either SNS or PNS demonstrates that the maximal frequency of the LF band is under the influences of both sympathetic and parasympathetic components [37, 43]. The LF/HF ratio is computed to estimate sympathovagal balance and is strongly correlated with the SD2/SD1 ratio [37, 39, 44]. In the present study, a transient increase in total power, which is the sum of VLF, LF, and HF power bands changed under the sympathetic and vagal influences [37, 45] and was detected, concurrent with increased SDNN at 30 minutes post-transport. This result could confirm the combined roles of sympathetic and vagal activities in horses soon after road transport ends. Despite the controversy over ANS involvement in the VLF power band [38, 39], a reduction in the VLF power band was observed in the present study; this showed a positive correlation with a change in LF/HF and SD2/SD1 ratio (S2–S4 Figs). This result may provide, in part, evidence of SNS influence on VLF modulation, similar to a previously described report [46].

Previous studies have addressed the increasing stress by observing decreased HRV during road transport [4, 5, 12, 47]. Surprisingly, in the present study, changes in HRV following the modulation of frequency domain variables were markedly contrasting with earlier studies, showing a transient increase in the HF power band, coinciding with a decreased LF power band during the first few hours of transport. Meanwhile, a transient decrease in LF/HF and SD2/SD1 ratios was also observed at 30 minutes into the journey, lasting 30 minutes and 120 minutes, respectively. Since increased HF power band and decreased LF/HF and SD2/SD1 ratios mirror predominant vagal activity [37, 38, 45], changes in the HRV variables mentioned above indicated increased vagal tone during the first half of the journey. It is plausible that the transport-experienced horses felt entirely familiar with these transport protocols and friendly

environments, resulting in no stress during transport in the present study. The implementation of different HRV variables, the degree of horses' experience, and the different transport conditions may be the underlying factors responsible for the discrepancy between the current experiment and former studies. Since RESP was modulated by the interaction effect of three independent variables, it was implied that different transport conditions caused distinct RESP modifications. However, the difference in RESP was observed only in ATF horses at 90 minutes post-transport, suggesting, in part, that the ATF condition exerted more impact. Notably, there was no detection of HRV modulation from time-domain methods during the transports, unlike the frequency domain. It is possible that the frequency-domain method was more sensitive than the time-domain method for assessing HRV modification in response to physical stimuli.

A correlation was observed between the internal truck environment and several HRV variables under different transport conditions. Concerning the correlation between parameters in the ATF and ATS conditions, RR intervals, HR, PNS index, and SNS index showed moderate correlations with the internal humidity and temperature in the ATF condition, while a strong correlation was detected in the ATS condition. More importantly, the correlation between the internal truck environment and HRV variables also differed between different loading conditions in the air-conditioned truck. In the ATF condition, the RR intervals and PNS index were negatively correlated with internal humidity but showed a positive correlation with temperature, while HR and SNS index displayed a positive correlation with humidity but a negative correlation with temperature. In contrast to the correlation in the ATF condition, RR intervals and PNS index in ATS condition showed a negative correlation with both internal humidity and temperature. In the meantime, HR and the SNS index were positively correlated with internal humidity and temperature. It is plausible that the moisture generated by a full load of horses was beyond the air conditioning system's capacity to maintain humidity levels. This situation resulted in a gradually increasing humidity within the closed system, even though the internal temperature remained low. On the contrary, the moisture produced by fewer horses in space loading may not overcome the air conditioning system's capacity to maintain internal humidity. With the same internal temperature, a distinctive internal humidity may be the factor causing a discrepancy in the correlations in the ATS in ATF conditions. These results suggest that different loading conditions impacted HRV modulation in air-conditioned trucks.

Although air-conditioning and loading conditions appeared to impact the modulation of cortisol levels and various HRV variables, the effect size of the interaction and main effects were small, leading to insufficient evidence on which transport method is the most appropriate for horses in tropical environments. This raises questions regarding whether significant evidence to conclusively determine the most appropriate transport method can be found in an experiment with a larger sample size. Furthermore, changes in hormonal and autonomic responses during long-distance transport in horses with differently conditioned trucks in tropical environments need further investigation.

The main limitation of this study was the small number of horses participating. Furthermore, fluctuation in cortisol levels cannot be determined during road transport because blood collections were inappropriate on trucks for fear of excitement, which could lead to unpredictable injuries and distort the HRV variable during the maneuver.

## Conclusion

Medium-distance road transport, irrespective of air-conditioning and loading conditions, caused no stress in transport-experienced horses in a tropical environment. Nonetheless, both conditions affected cortisol levels, HR, and HRV modulation, leading horses to respond in

distinctive ways to differently conditioned trucks during road transport. In contrast, a transient increase in vagal activity suggests relaxation during the first hours of transport. Even though medium-distance road transport caused no stress in horses in the present study, stress responses may vary in some tropical regions due to the dynamic of weather conditions in different seasons. Nevertheless, these results provided fundamental information regarding the stress responses in horses undergoing transport in differently conditioned trucks. At least in part, this information aids in road transport management and suggests the need for further research concerning horse welfare during road transport in tropical environments.

## Supporting information

**S1 Fig. Correlation among inside humidity, inside temperature and HRV variables in ATF horses during road transport. ATF**; air-conditioned truck with a full load, **RR**; beat-to-beat interval, **HR**; heart rate, **SDNN**; the standard deviation of the normal-to-normal beat interval, **RMSSD**; Square root of the mean squared differences between successive RR intervals, **pNN50**; the relative number of the beat-to-beat intervals that differs than 50 ms, **TINN**; triangular interpolation of normal-to-normal intervals, **RRTI**; RR triangular index, **VLF**; very-low-frequency band, **LF**; low-frequency band, **HF**; high-frequency band, **SD1**; the standard deviation of Poincaré plot perpendicular to the line-of-identity, **SD2**; the standard deviation of Poincaré plot along the line-of-identity, **PNS**; the parasympathetic nervous system and **SNS**; the sympathetic nervous system.
(DOCX)

**S2 Fig. Correlation among inside humidity, inside temperature and HRV variables in ATS horses during road transport. ATS**; air-conditioned truck with a space load, **RR**; beat-to-beat interval, **HR**; heart rate, **SDNN**; the standard deviation of the normal-to-normal beat interval, **RMSSD**; Square root of the mean squared differences between successive RR intervals, **pNN50**; the relative number of the beat-to-beat intervals that differs than 50 ms, **TINN**; triangular interpolation of normal-to-normal intervals, **RRTI**; RR triangular index, **VLF**; very-low-frequency band, **LF**; low-frequency band, **HF**; high-frequency band, **SD1**; the standard deviation of Poincaré plot perpendicular to the line-of-identity, **SD2**; the standard deviation of Poincaré plot along the line-of-identity, **PNS**; the parasympathetic nervous system and **SNS**; the sympathetic nervous system.
(DOCX)

**S3 Fig. Correlation among inside humidity, inside temperature and HRV variables in N-ATF horses during road transport. ATS**; air-conditioned truck with a space load, **RR**; beat-to-beat interval, **HR**; heart rate, **SDNN**; the standard deviation of the normal-to-normal beat interval, **RMSSD**; Square root of the mean squared differences between successive RR intervals, **pNN50**; the relative number of the beat-to-beat intervals that differs than 50 ms, **TINN**; triangular interpolation of normal-to-normal intervals, **RRTI**; RR triangular index, **VLF**; very-low-frequency band, **LF**; low-frequency band, **HF**; high-frequency band, **SD1**; the standard deviation of Poincaré plot perpendicular to the line-of-identity, **SD2**; the standard deviation of Poincaré plot along the line-of-identity, **PNS**; the parasympathetic nervous system and **SNS**; the sympathetic nervous system.
(DOCX)

**S4 Fig. Correlation among inside humidity, inside temperature and HRV variables in N-ATS horses during road transport. ATS**; air-conditioned truck with a space load, **RR**; beat-to-beat interval, **HR**; heart rate, **SDNN**; the standard deviation of the normal-to-normal beat interval, **RMSSD**; Square root of the mean squared differences between successive RR

intervals, **pNN50**; the relative number of the beat-to-beat intervals that differs than 50 ms, **TINN**; triangular interpolation of normal-to-normal intervals, **RRTI**; RR triangular index, **VLF**; very-low-frequency band, **LF**; low-frequency band, **HF**; high-frequency band, **SD1**; the standard deviation of Poincaré plot perpendicular to the line-of-identity, **SD2**; the standard deviation of Poincaré plot along the line-of-identity, **PNS**; the parasympathetic nervous system and **SNS**; the sympathetic nervous system.
(DOCX)

## Acknowledgments

The authors would like to thank the Bureau of Disease Control and Veterinary Service, Department of Livestock Development, for granting permission to conduct the horse movement in this study. We are also grateful to Vaewratt Kamonkon and Sailub Lertratanachai for allocating their horses for this study. In this study, Thai Polo and Equestrian Club and the Horse Lover's Club are thanked for providing non-air-conditioned and air-conditioned trucks for horse movement. The determination of cortisol levels by Thai Vet Lab Co., Ltd. is greatly appreciated.

## Author Contributions

**Conceptualization:** Siengsaw Lertratanachai, Chanoknun Poochipakorn, Kanokpan Sanigavatee, Thita Wonghanchao, Metha Chanda.

**Data curation:** Siengsaw Lertratanachai, Chanoknun Poochipakorn, Kanokpan Sanigavatee, Thita Wonghanchao, Metha Chanda.

**Formal analysis:** Siengsaw Lertratanachai, Chanoknun Poochipakorn, Kanokpan Sanigavatee, Onjira Huangsaksri, Thita Wonghanchao, Ponlakrit Charoenchanikran, Chaipat Lawsirirat, Metha Chanda.

**Funding acquisition:** Metha Chanda.

**Investigation:** Chanoknun Poochipakorn, Kanokpan Sanigavatee, Onjira Huangsaksri, Thita Wonghanchao, Ponlakrit Charoenchanikran, Chaipat Lawsirirat, Metha Chanda.

**Methodology:** Siengsaw Lertratanachai, Chanoknun Poochipakorn, Kanokpan Sanigavatee, Onjira Huangsaksri, Thita Wonghanchao, Ponlakrit Charoenchanikran, Chaipat Lawsirirat, Metha Chanda.

**Project administration:** Metha Chanda.

**Resources:** Siengsaw Lertratanachai, Metha Chanda.

**Software:** Chanoknun Poochipakorn, Kanokpan Sanigavatee, Thita Wonghanchao, Metha Chanda.

**Supervision:** Chanoknun Poochipakorn, Kanokpan Sanigavatee, Thita Wonghanchao, Metha Chanda.

**Validation:** Chanoknun Poochipakorn, Onjira Huangsaksri, Ponlakrit Charoenchanikran, Chaipat Lawsirirat, Metha Chanda.

**Visualization:** Chanoknun Poochipakorn, Kanokpan Sanigavatee, Thita Wonghanchao, Metha Chanda.

**Writing – original draft:** Siengsaw Lertratanachai, Chanoknun Poochipakorn, Kanokpan Sanigavatee, Onjira Huangsaksri, Thita Wonghanchao, Ponlakrit Charoenchanikran, Chaipat Lawsirirat, Metha Chanda.

**Writing – review & editing:** Metha Chanda.

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
