## [Decision Letter · Decision Letter 0]

5 Aug 2024

PONE-D-24-11922Cortisol levels, heart rate, and autonomic responses in horses during repeated road transport with different truck conditions trucks in a tropical environmentPLOS ONE

Dear Dr. Chanda,

Thank you for submitting your manuscript to PLOS ONE. After careful consideration, we feel that it has merit but does not fully meet PLOS ONE’s publication criteria as it currently stands. Therefore, we invite you to submit a revised version of the manuscript that addresses the points raised during the review process.

We look forward to receiving your revised manuscript.

Kind regards,

Chris Rogers

Academic Editor

PLOS ONE

3. We note that Fig S1 in your submission contain [map/satellite] images which may be copyrighted. All PLOS content is published under the Creative Commons Attribution License (CC BY 4.0), which means that the manuscript, images, and Supporting Information files will be freely available online, and any third party is permitted to access, download, copy, distribute, and use these materials in any way, even commercially, with proper attribution. For these reasons, we cannot publish previously copyrighted maps or satellite images created using proprietary data, such as Google software (Google Maps, Street View, and Earth). For more information, see our copyright guidelines: http://journals.plos.org/plosone/s/licenses-and-copyright.

1. You may seek permission from the original copyright holder of Fig S1 to publish the content specifically under the CC BY 4.0 license. 

Additional Editor Comments:

Apologies for the delay with this manuscript. We have now secured the number of reviews and i am happy to say it may proceed with the publication process subject to minor revision.

Reviewers' comments:

Reviewer's Responses to Questions

**Comments to the Author**

1. Is the manuscript technically sound, and do the data support the conclusions?

Reviewer #1: Partly

Reviewer #2: Yes

Reviewer #3: Yes

2. Has the statistical analysis been performed appropriately and rigorously? 

Reviewer #1: N/A

Reviewer #2: Yes

Reviewer #3: Yes

3. Have the authors made all data underlying the findings in their manuscript fully available?

Reviewer #1: Yes

Reviewer #2: Yes

Reviewer #3: Yes

4. Is the manuscript presented in an intelligible fashion and written in standard English?

Reviewer #1: Yes

Reviewer #2: Yes

Reviewer #3: Yes

5. Review Comments to the Author

Reviewer #1: This is a very interesting area of study that is important for ensuring best welfare standards in equine transportation management. The manuscript is well written, and uses a comprehensive set of stress response measures nicely, well done. However, it is regrettable that on this occasion the sample size is too small to draw any meaningful conclusions.

I look forward to seeing further research investigating this topic, studied in a larger cohort of horses.

Reviewer #2: This was an interesting article. Thorough analysis and well presented. I only had a couple of minor comments

Line 21 use temperate rather than cold climates here

Line 22 did you estimate or did you measure these levels / parameters ?

Line 44 number of shows seems low – are you just retrieving data from one discipline – the 150 shows is only 3 per weekend – in Show Jumping alone there are at least 6 to 9 FEI shows each weekend – please check numbers or change text

Line 72 temperate rather than cold climates or non-tropical may be better descriptors

Table 1 mean drive speed as assume this speed was not constant throughout trial.

For materials and methods or similar Given this was travel on a highway is it possible to somehow describe the characteristics of the route ie took 10 mins from loading site until on highway and able to maintain a steady state ? or similar

Reviewer #3: This paper is interesting and provides a good approach to examining the stress responses to trucking conditions. The discussion explains the results with good comparison to other literature

A few minor comments

Please edit p-values so that the p is lowercase and all values are rounded to 3 decimal places.

Table 1 and 3 could do with a bit more information in the title to make it standalone

Were windows closed during transport?

6. PLOS authors have the option to publish the peer review history of their article (what does this mean?). If published, this will include your full peer review and any attached files.

Reviewer #1: No

Reviewer #2: No

Reviewer #3: No

---

## [Author Response · Author response to Decision Letter 0]

21 Aug 2024

Response to the editor and reviewers

Title: Cortisol levels, heart rate, and autonomic responses in horses during repeated road transport with differently conditioned trucks in a tropical environment

Dear Editor and Reviewers,

We’d like to thank you for dedicated time to reviewer our work and give valuable suggestions and comments to improve this manuscript. We’ve addressed all point raised by the editor and reviewers. The corrections are highlighted in green while the statements that have already been mentioned are highlighted in yellow.

Response to the editor

We’ve revised the content and headings according to the journal-style templates.

Response to the editor

We’ve added the verbal consent and ethical approval statement on pages 4-5, lines 88-90.

3. We note that Fig S1 in your submission contain [map/satellite] images which may be copyrighted. All PLOS content is published under the Creative Commons Attribution License (CC BY 4.0), which means that the manuscript, images, and Supporting Information files will be freely available online, and any third party is permitted to access, download, copy, distribute, and use these materials in any way, even commercially, with proper attribution. For these reasons, we cannot publish previously copyrighted maps or satellite images created using proprietary data, such as Google software (Google Maps, Street View, and Earth). For more information, see our copyright guidelines: http://journals.plos.org/plosone/s/licenses-and-copyright. We require you to either (1) present written permission from the copyright holder to publish these figures specifically under the CC BY 4.0 license, or (2) remove the figures from your submission:

Response to the editor

We decided to remove the S1 Fig from the manuscript to avoid breaching the copyright.

 Response to the editor

We’ve modified a bit in the reference list by adding the DOI in all journal article to ease the reader tracking those cited article.

 

Review Comments to the Author

Reviewer #1: This is a very interesting area of study that is important for ensuring best welfare standards in equine transportation management. The manuscript is well written, and uses a comprehensive set of stress response measures nicely, well done. However, it is regrettable that on this occasion the sample size is too small to draw any meaningful conclusions.

I look forward to seeing further research investigating this topic, studied in a larger cohort of horses.

Response to the reviewer.

We’d like to thank the reviewer for this kind word. We agree that the sample size is too small, and we have addressed this as the limitation of this study.

Reviewer #2: This was an interesting article. Thorough analysis and well presented. I only had a couple of minor comments

Line 21 use temperate rather than cold climates here

Response to the reviewer.

We’ve revised it on page 2, line 22 accordingly.

Line 22 did you estimate or did you measure these levels / parameters ?

Response to the reviewer.

We’ve revised it by using measured instead of estimate on page 2, line 23 accordingly.

Line 44 number of shows seems low – are you just retrieving data from one discipline – the 150 shows is only 3 per weekend – in Show Jumping alone there are at least 6 to 9 FEI shows each weekend – please check numbers or change text

Response to the reviewer.

We’d like to thank the reviewer for this notification. We’ve given wrong information. In fact, there are over 3200 shows organizing annually regarding the FEI database. We’ve revised it on page 3, line 45 accordingly.

Line 72 temperate rather than cold climates or non-tropical may be better descriptors

Table 1 mean drive speed as assume this speed was not constant throughout trial.

Response to the reviewer.

We’ve revised to use the temperate instead of the cold environment on page 4, line 73, accordingly. Regarding the driving speed, the driver has been constructed to drive approximately 90 km/h according to the speed gadget of the truck. However, they can not keep this speed constant throughout the trip as the speed has sometimes been slowed down due to transient traffic jams or waiting to pay a toll fee. The mean driving speed was computed from the polar flow program with respect to the data from one sports watch placed within the truck of each trip.

For materials and methods or similar Given this was travel on a highway is it possible to somehow describe the characteristics of the route ie took 10 mins from loading site until on highway and able to maintain a steady state ? or similar

Response to the reviewer.

We agree with the reviewer that describing the characteristics of the route may provide clear information to the reader. However, based on the response to the reviewer above, the driver could not keep the speed of 90 km/h as we instructed throughout the trip due to the transient traffic jam or waiting to pay the toll fee. These factors resulted in different driving speeds between the dates of the experiment, even though the trucks were driven on the same route. So, in our opinion, describing the characteristics of the route may confuse the reader rather than give them useful information. Please correct me if I am wrong.

Reviewer #3: This paper is interesting and provides a good approach to examining the stress responses to trucking conditions. The discussion explains the results with good comparison to other literature

A few minor comments

Please edit p-values so that the p is lowercase and all values are rounded to 3 decimal places.

Table 1 and 3 could do with a bit more information in the title to make it standalone

Were windows closed during transport?

Response to the reviewer.

We’ve revised all p-vales in lowercase throughout the text accordingly. In a non-air-conditioned truck, the windows were open throughout the trips.

---

## [Editor Report · Decision Letter 1]

26 Aug 2024

Cortisol levels, heart rate, and autonomic responses in horses during repeated road transport with differently conditioned trucks in a tropical environment

PONE-D-24-11922R1

Dear Dr. Chanda,

We’re pleased to inform you that your manuscript has been judged scientifically suitable for publication and will be formally accepted for publication once it meets all outstanding technical requirements.

Kind regards,

Chris Rogers

Academic Editor

PLOS ONE

Additional Editor Comments (optional):

Thank you for the edits to the manuscript.
---

## [Editor Report · Acceptance letter]

29 Aug 2024

PONE-D-24-11922R1 

PLOS ONE

Dear Dr. Chanda, 

I'm pleased to inform you that your manuscript has been deemed suitable for publication in PLOS ONE. Congratulations! Your manuscript is now being handed over to our production team.

Kind regards, 

on behalf of

Dr. Chris Rogers 

Academic Editor

PLOS ONE